# Monoclonal Gammopathy of Undetermined Significance (MGUS)—Not So Asymptomatic after All

**DOI:** 10.3390/cancers12061554

**Published:** 2020-06-12

**Authors:** Oliver C. Lomas, Tarek H. Mouhieddine, Sabrin Tahri, Irene M. Ghobrial

**Affiliations:** 1Department of Medical Oncology, Dana-Farber Cancer Institute, Boston, MA 02115, USA; OliverC_Lomas@dfci.harvard.edu (O.C.L.); TarekH_Mouhieddine@DFCI.HARVARD.EDU (T.H.M.); Sabrin_tahri@dfci.harvard.edu (S.T.); 2Department of Medicine, Icahn School of Medicine at Mount Sinai, New York, NY 10029, USA

**Keywords:** MGUS, multiple myeloma, paraprotein, neuropathy, fracture, MGRS, osteoporosis

## Abstract

Monoclonal Gammopathy of Undetermined Significance (MGUS) is considered to be a benign precursor condition that may progress to a lymphoproliferative disease or multiple myeloma. Most patients do not progress to an overt condition, but nevertheless, MGUS is associated with a shortened life expectancy and, in a minority of cases, a number of co-morbid conditions that include an increased fracture risk, renal impairment, peripheral neuropathy, secondary immunodeficiency, and cardiovascular disease. This review aims to consolidate current evidence for the significance of these co-morbidities before considering how best to approach these symptoms and signs, which are often encountered in primary care or within a number of specialties in secondary care.

## 1. Introduction

Monoclonal Gammopathy of Undetermined Significance (MGUS) is characterized by the presence of a serum monoclonal paraprotein derived from immunoglobulin (Ig). MGUS may be classified into IgM and non-IgM MGUS, depending on the cellular clone responsible for the particular paraprotein. In most cases, IgM MGUS may develop into lymphoid malignancies, especially Waldenström’s macroglobulinemia (WM), but also, rarely, other non-Hodgkin lymphomas such as chronic lymphocytic leukemia [1]. Non-IgM MGUS is derived from mature plasma cells that may progress to multiple myeloma (MM) [2]. In a minority of cases, MGUS can be identified as light chain only, which refers to the isolated secretion of κ or λ light chains of immunoglobulin [3]. Light chains derived from all variants of MGUS may aggregate in and impair organs such as the kidney and heart. Depending on the nature of light chain infiltration, amyloid light chain (AL) amyloidosis or light-chain deposition disease may be the pathophysiological process involved. MGUS is found in approximately 3% of the population over the age of 50 years, with a median age of presentation of 72 years [4]. In addition to increasing age, non-IgM MGUS is found more frequently in men and in Afro-Caribbean compared to Caucasian populations [5]. Even though the vast majority of the cases of MM arise from a prior state of MGUS [6], overall, the annual risk of progression from MGUS to symptomatic MM, WM, or other related disorders is only ∼1% [2]. Therefore, most patients with MGUS do not progress to symptomatic MM or other lymphoproliferative disorders [7]. However, even in the absence of malignancy and when matched by age and sex, MGUS patients experience shorter overall survival from diagnosis than is expected (8.1 vs. 12.4 years) according to a prospective cohort study [2]. In addition to progression to lymphoproliferative diseases and amyloidosis, patients with MGUS appear to suffer from a greater prevalence of recurrent infections, ischemic heart disease, peripheral neuropathy, and renal diseases compared to those without MGUS [8]. Cutaneous manifestations of monoclonal gammopathies have also been described, which are reviewed comprehensively elsewhere [9]. These co-morbidities afflict only a minority of patients with MGUS, but nevertheless they need to be considered on an individual patient basis. Since the prevalence of MGUS is much higher than that of MM, these more moderate symptoms that may be associated with MGUS may nevertheless, have a significant impact on healthcare services. MGUS may represent an under-appreciated pathological state with attributable morbidity and mortality different from progression to malignancy.

The aim of this review is to examine the most consistently reported co-morbidities associated with MGUS, namely the increased risk of bone fractures, peripheral neuropathy, renal impairment, secondary immunodeficiency, and cardiovascular disease [10]. A summary of the diagnostic criteria for MGUS is shown in Table 1.

## 2. Osteoporosis and Bone Fractures

While lytic bone lesions are a defining feature of symptomatic MM [11], a number of population-based studies have demonstrated an association between MGUS and low bone mineral density/osteoporosis, which, in turn, increases the risk of fractures within, or close to the axial skeleton [10]. The location of the fractures reflects classical sites of involvement by myeloma, suggesting shared pathophysiology, and may be related to elevated serum receptor activator of nuclear factor k-Β ligand/osteoprotegerin (RANK-L/OPG) ratios [12]. The occurrence of a non-traumatic fracture in the context of MGUS requires careful assessment by a hematologist with access to cross-sectional imaging and a bone marrow biopsy to exclude a myeloma-defining event. Furthermore, the occurrence of these fractures in the context of MGUS does not herald progression to MM [13].

Further evidence of an increased risk of fractures is supported by a comorbidity-adjusted Danish population cohort study (adjusted relative risk (RR) of 1.4) [14], as well as a Swedish registry study that confirmed a higher risk (HR) for fractures of the axial skeleton in MGUS patients (HR 2.37 for vertebral/pelvis fractures) [13]. These observations have been supported by a meta-analysis of 60,000 individuals, which reported that patients with MGUS are at higher risk of suffering vertebral fractures, in particular, compared to controls, with a RR of 2.50 [15]. Among patients referred to an osteoporosis clinic, MGUS was diagnosed in 3.6% of those with osteoporosis and 2% of those with normal bone mineral density [16]. Studies have attempted to identify characteristics within the MGUS population that predict higher fracture risk. Older age, rather than sex, is correlated with increased risk of fractures [17]. The serum concentration of the monoclonal paraprotein does not correlate with fracture risk, but the class of the immunoglobulin may be significant. The IgA paraprotein subtype has been shown to predispose to fractures, but there are conflicting reports as to whether fracture risk relates to either κ or λ light chain excess [10,17,18].

Dual Energy X-ray Absorptiometry (DEXA) scans are recommended by the International Myeloma Working Group (IMWG) to detect low bone mineral density in MGUS [19]. In a study of MGUS patients with fractures compared to those without fractures, the bone mineral density of the hip and lumbar spine was found to be lower in those with fractures [20]. The use of bisphosphonates in patients with MGUS with reduced bone mineral density is associated with a reduced risk of fractures [21]. In the absence of further risk stratification for fractures in the context of MGUS, oral bisphosphonates are indicated when osteoporosis is identified [19]. The European Myeloma Network suggests that for patients with reduced bone mineral density or a history of non-traumatic fracture, bisphosphonate therapy, as well as replacement of calcium and vitamin D, are indicated [22]. Bisphosphonate therapy may be in the form of weekly oral alendronate [21] or six-monthly intravenous zoledronic acid [23].

However, differences in bone mineral density between patients with MGUS and controls have not been consistently demonstrated prospectively. Analysis of bone mineral density and bone geometry by computed tomography (CT) scans in more than 5000 individuals over almost seven years has suggested that the fracture risk in MGUS patients was associated with increased bone volume due to changes in bone geometry rather than decreased bone mineral density [24]. Prospective studies using novel biomarkers such as volumetric CT scans and clinically relevant end-points of fracture risks are required.

## 3. Monoclonal Gammopathy of Renal Significance (MGRS)

The term monoclonal gammopathy of renal significance (MGRS) was first introduced in 2012 by the International Kidney and Monoclonal Gammopathy Research Group (IKMG) to describe a set of renal disorders that were characterized by the deposition of monoclonal immunoglobulin in the kidney [11,25,26]. MGRS currently encompasses all precursor B-cell or plasma cell clonal disorders that secrete a nephrotoxic monoclonal paraprotein. Unlike non-monoclonal gammopathy renal diseases, such as IgA or membranous nephropathy, MGRS is usually isolated and progressive [27]. Once MGRS progresses to overt MM or lymphoma, patients no longer satisfy the MGRS criteria and should be managed as per their respective disease-specific protocols.

Typically, patients present with unexplained renal impairment or proteinuria, and the diagnosis of MGRS is confirmed by renal biopsy that detects monoclonal immunoglobulin deposits in the glomeruli, tubules, vessels, or interstitium of the kidney. MGRS is categorized by glomerulopathies in either the presence or absence of monoclonal immunoglobulin deposition. Among the glomerulopathies with immunoglobulin deposits, there are the organized immunoglobulin depositions, such as fibrillar (immunoglobulin light chain (AL), immunoglobulin heavy chain (AH), immunoglobulin light and heavy chain (ALH), and monoclonal fibrillary glomerulonephritis), microtubular (type I and type II cryoglobulinemia, immunotactoid glomerulopathy (ITG)) and the inclusion of crystalline deposits (light-chain proximal tubulopathy (LCPT), crystal-storing histiocytosis, and cryocrystalglobulin glomerulonephritis) [28,29]. Furthermore, the disorders of non-organized deposits include monoclonal immunoglobulin deposition diseases (MIDD) and proliferative glomerulonephritis with monoclonal immunoglobulin deposits (PGNMID) and tubular disorders such as Fanconi syndrome [28,29].

Currently, only C3 glomerulopathy is recognized as a disorder associated with MGRS that has no monoclonal immunoglobulin deposition. C3 glomerulopathy has been found to frequently coexist with MGUS [30,31,32,33], whereby up to 65.1% of C3 glomerulopathy patients above the age of 50 years could concurrently have MGUS [34]. While the role of MGUS in the pathogenesis of C3 glomerulopathy remains to be elucidated, it is hypothesized that the monoclonal immunoglobulins may be impairing the regulation of the complement alternative pathway [35]. Moreover, thrombotic microangiopathy has been tentatively added to the list of MGRS with absent immunoglobulin deposition disorders as more studies investigate the prevalence of its coexistence with MGUS that seem to reach up to 21% in those who are 50 years or older [36,37], and seems to be due to an autoantibody effect of the monoclonal protein against a complement regulatory protein [38]. Finally, polyneuropathy, organomegaly, endocrinopathy, monoclonal gammopathy, and skin changes (POEMS) syndrome also seems to be associated with glomerular nephropathy that is probably due to a cytokine-mediated endothelial injury [39].

The most serious complication of MGRS is end-stage renal disease (ESRD) and the comorbidities it entails. In a retrospective study of 37 patients with MGRS, it was found that 22% of patients eventually progressed to ESRD at a mean follow-up of 30.3 months [40]. Another retrospective evaluation of 19 MIDD patients, found that the five-year ESRD-free survival was only 37% [41]. MGUS patients who present with renal impairment should be evaluated for AL-amyloidosis and other causes of renal dysfunction. A high index of suspicion for MGRS should be maintained and, unless there is a contraindication, a renal biopsy should be obtained in patients with unexplained renal impairment and monoclonal gammopathy. As in the investigation of suspected MM, bone marrow biopsy and cross-sectional imaging should be performed at baseline to identify the nature of the disease and if there is an extra-medullary distribution of the disease. A solitary plasmacytoma may harbor the plasma cell clone responsible for the nephropathy. The identification of such a plasmacytoma changes management as isolated radiotherapy may be curative and hence limits renal impairment [42]. Once the diagnosis is confirmed, close co-operation with nephrologists is advised to determine the optimum therapeutic intervention based on the type of MGRS, degree of renal impairment, and risk of progression to ESRD.

In the absence of strategies to block the deposition of the pathogenic protein, the principal therapeutic target is the underlying B-cell clone that secretes it. Treatment options usually involve the use of chemotherapies (melphalan, cyclophosphamide, and bendamustine), dexamethasone, bortezomib, or immunotherapies like rituximab, employed with the goal of preserving renal function and targeting the clonal population of cells that are producing the monoclonal immunoglobulin [25,43,44,45,46,47]. In the context of renal impairment, certain drugs are preferred, such as cyclophosphamide and bortezomib that are better tolerated than melphalan and lenalidomide. Despite a limited response to melphalan and prednisone in light-chain deposition disease [48], a case series of four dialysis-dependent MIDD patients reported that high-dose melphalan followed by autologous stem cell transplant to be a safe and effective option that resulted in durable responses and subsequently allowed patients to receive a kidney transplant [49]. Many patients who develop end-stage renal impairment are often not considered for kidney transplantation due to their high rate of recurrence, as eradication of the underlying B-cell clone is not frequently possible [36,37,38]. However, autologous stem cell transplantation should still be considered as in the case of MM [50].

The wide spectrum of MGRS combined with the high prevalence of MGUS [4] and chronic kidney disease, which in turn may be due to other medical conditions such as hypertension and diabetes mellitus [51], may complicate and delay the accurate diagnosis of MGRS. Thus, it is imperative that a concurrent hematologic and renal evaluation be performed in those who are suspected of having MGRS.

## 4. Peripheral Neuropathy

Peripheral neuropathy describes the impairment of somatic (motor and sensory), enteric or autonomic neurons outside the central nervous system of the brain and spinal cord. There are many causes of peripheral neuropathy from systemic disorders, such as diabetes mellitus and nutrient deficiencies, to drug side-effects, as well as neuroanatomical causes such as vertebral radiculopathy. Furthermore, sensorimotor or autonomic impairment as part of peripheral neuropathy may contribute to falls, which increases the risk of fractures, thereby increasing the risk of fractures to compound the co-morbidities associated with MGUS.

The association between monoclonal paraproteinemia and neuropathy has been demonstrated in a population-based screening study, which found a higher than expected prevalence of peripheral neuropathy [10]. A retrospective analysis of a large Scandinavian cohort has found an approximately three-fold increase in the relative risk for peripheral neuropathy in cases of MGUS, independently of diabetes mellitus [52]. The absolute prevalence of peripheral neuropathy appears to be under 5% of MGUS patients [53]. Apart from the identification of the paraprotein, the onset, duration, and nature of the neuropathy must be carefully examined. Nerve conduction studies often help to identify objectively whether there is demyelinating or axonal neurological injury prior to considering therapy. An algorithmic approach to the initial investigation of neuropathy in association with a monoclonal paraprotein is shown in Figure 1.

IgM is the most commonly associated paraprotein with peripheral neuropathy, compared to IgG and IgA, at a ratio of approximately 6:3:1 [54]. The IgM paraprotein appears to have a direct effect on peripheral nerves, leading to loss of the protective myelin sheath of the nerve (demyelination), which impairs nerve conduction velocity. In approximately 50% of patients with IgM MGUS, the paraprotein binds myelin-associated glycoprotein (MAG), while in the remainder of cases, the reactivity of the paraprotein is found against ganglioside or asialo-GM1 [55,56]. The clinical presentation of distal, demyelinating, symmetric neuropathy similar to Chronic Inflammatory Demyelinating Polyneuropathy (CIDP) is found irrespective of the associated antibody. In the case of anti-MAG antibodies, higher titers are associated with the development of symptoms [57]. IgM MGUS is derived from a CD19+/CD20+ lymphoid population for which the anti-CD20 monoclonal antibody therapy, rituximab, has been used successfully. A prospective single-arm trial of ten patients with anti-MAG IgM-related neuropathy found all patients to have improved after 375 mg/m^2^ rituximab monotherapy every four weeks for twelve months [58]. However, the response was maintained in only six patients at three years post-rituximab monotherapy. A randomized, placebo-controlled trial of rituximab in IgM MGUS patients with neuropathy and anti-MAG antibodies showed benefit in one-third of the treatment arm compared to stable disease or worsening symptoms in the placebo arm [59]. Rituximab treatment is usually associated with a decrease in the titer of IgM autoantibody [60]. If the symptoms are sufficiently severe, therapies such as those that are used in WM may be considered, although patients must be counseled very carefully about the risks of therapy, which may prove to be more difficult than the presenting neuropathy.

The association of non-IgM MGUS and peripheral neuropathy first needs consideration of the lymphoproliferative disorders with distinct management of AL amyloidosis and POEMS syndrome (an acronym of Polyneuropathy, Organomegaly, Endocrinopathy, Monoclonal gammopathy, and Skin changes). POEMS is a monoclonal plasma disorder that is characterized by a progressive, chronic, demyelinating sensorimotor polyneuropathy, which is mandatory for the diagnosis of the condition [61]. The neuropathy displays a motor tendency and is often found at presentation. The associated monoclonal paraprotein is usually IgG or IgA, and the bone lesions are sclerotic rather than lytic as in MM. Therapy involves chemoradiotherapy and often autologous stem cell transplantation [61,62]. AL amyloidosis describes the deposition of immunoglobulin (whole or light-chain) in β-pleated sheets in target organs such as the kidneys and heart. All forms of MGUS can progress to AL amyloidosis, with λ light chains being the predominant light chain class involved [63].

Peripheral neuropathy affects up to 20% of patients with AL amyloidosis. A large cohort study from Scandinavia has found that peripheral neuropathy was associated with an almost three-fold increase in the risk of light chain amyloidosis. [52]. The neuropathy of amyloidosis frequently presents with painful sensory symptoms or autonomic dysfunction, and nerve conduction studies often reveal an axonal pattern that helps to differentiate from the demyelinating phenotype observed in IgM-MGUS neuropathy. Compared to other neuropathies associated with monoclonal proteins, the clinical course of neuropathy in AL amyloidosis is usually more progressively debilitating [63]. Similar to renal AL amyloidosis, systemic chemotherapy therapy with bortezomib, cyclophosphamide, and dose-attenuated lenalidomide are frequently used therapies in this multi-system disorder.

Given that the prevalence of MGUS is 3% in people over the age of 50, distinguishing causality from a correlation of MGUS to peripheral neuropathy is a diagnostic challenge, as the presence of a monoclonal protein in a patient with neuropathy does not necessarily mean that they are related. Patients with MGUS and evidence of neuropathy should undergo an extended evaluation to rule out a hematologic disease, including a fat pad biopsy to look for amyloidosis and screening for anti-ganglioside and anti-MAG antibodies, as well as cryoglobulins. Furthermore, collaboration with a neurologist is recommended for these patients to obtain an electromyogram or neuromuscular testing and their interpretation.

## 5. Immunodeficiency in MGUS

MGUS is a recognized cause of secondary antibody deficiency. The prevalence of MGUS increases with age [2], as does an age-related decline in immune function, or immunosenescence [64]. However, an age and sex-matched retrospective cohort study of over 5,000 MGUS patients revealed a two to three-fold increase in the relative risk of bacterial and viral infections in MGUS compared to controls [65]. The increase in risk was not influenced by the paraprotein isotype but was positively correlated with the M-protein concentration. Unfortunately, the study failed to provide quantitative data describing the background immunoglobulins, and, therefore, whether the increase in the risk was correlated to hypogammaglobulinemia is not known. Hypogammaglobulinemia describes the decrease in at least one of the non-monoclonal immunoglobulin heavy chain classes. Hypogammaglobulinemia is present in approximately 30% of patients with MGUS [4] and rises to 70% in smoldering multiple myeloma (SMM) [66]. Indeed, in SMM, polyclonal hypogammaglobulinemia had a median rate of progression to myeloma of 68% at five years compared to 38% with monoclonal hypogammaglobulinemia [66]. Impairment of humoral and cellular immunity has been implicated in the susceptibility to infection of patients, especially with newly diagnosed MM [67]. Analysis of the United Kingdom Medical Research Council’s Myeloma trials from 1982 to 2002 revealed that almost half of all deaths were attributed to infections such as pneumonia during the first three months after diagnosis [68]. Induction therapy itself may have been immunosuppressive and, thus contributing to this susceptibility. However, the risk of infection decreased with the response to therapy, which suggests some immune reconstitution accompanies a lower burden of myeloma. In addition to progressive impairment of humoral immunity, changes across many cell types of the bone marrow immune microenvironment, including Natural Killer (NK), Dendritic, and T-cells, have been observed from MGUS, to SMM, and MM [69]. These observations support the concept that there is progressive immunoparesis that accompanies disease progression from MGUS to MM. Whether this loss of immune capacity permits myeloma progression, is a consequence of it, or perhaps a combination of both is not well understood.

A history of, particularly sinopulmonary, infections should be sought in patients with MGUS, and for whom associated hypogammaglobulinemia may be contributory. Laboratory assessment comprises serum concentrations of IgG, IgA, and IgM, as well as specific antibody responses to tetanus, pneumococcus, and Hemophilus. In this way, both the quantity and quality of humoral immunity are assessed. Not all patients with hypogammaglobulinemia present with recurrent infections, and conversely, serious infections may occur with normal immunoglobulin concentrations due to other reasons such as altered T-cell or NK cell function [70]. By comparison of antibody titers to specific bacterial and fungal pathogens, humoral immunity has been found to be depressed in MGUS, and to a lesser degree compared to MM [71]. Usually, in consultation with immunologists, a trial of prophylactic antibiotics or immunoglobulin replacement may be trialed over a sufficiently long time (at least a year) to assess the benefit of the intervention across seasonal variations in infections.

The majority of patients with MGUS are over the age of 65 [4], and even in healthy individuals, the immune response to vaccination declines with age [72]. In particular, MGUS patients have a lower than anticipated antibody responses to influenza and Hepatitis B vaccines [73,74]. Such immunosenescence may be mitigated by certain vaccination strategies. The administration of a dual high-dose influenza vaccine in MM patients may lead to more protective humoral responses [75]. While a 13-valent pneumococcal conjugate vaccine (PCV13) vaccination effectively protects MGUS patients against *Streptococcus pneumonia*, it was found to be more effective in patients with a lower concentration of monoclonal paraprotein [76]. A vaccine against a broader range of serotypes, a 23-valent pneumococcal polysaccharide vaccine (PPSV23), is currently licensed for use in the elderly [77]. In the context of MM, vaccination with PCV13 and PPSV23 was associated with significant increases in anti-pneumococcal antibodies in 85% of patients, independent of the degree of hypogammaglobulinemia [78]. These observations have not been replicated in MGUS but may be reasonably interpolated considering the lower burden of disease and degree of hypogammaglobulinemia in precursors states.

Changes in antibody titer following vaccination are a common method of assessing the immune response to vaccination [79]. However, this may need to be complemented by a functional antibody assay as antibody titers alone may give a misleading sense of immune response, especially with more severe dysfunction of humoral immunity, as is seen as MGUS progresses to MM [80]. Therefore, an approach to take with patients with MGUS may be to identify those patients with deficient antibody responses to vaccine antigens such as non-conjugated and conjugated pneumococcal; tetanus, diphtheria, meningococcus, and *Haemophilus influenzae B* [81]. From the observations that the response to vaccination is negatively correlated with disease progression [73], vaccination early in the disease would be expected to elicit greater antibody responses compared to those administered during later stages of the disease. This has been proposed in another lymphoid malignancy: Chronic Lymphocytic Leukemia (CLL), and its precursor state, Monoclonal B-cell Lymphocytosis (MBL) [82]. However, such an approach would benefit from rigorous testing against a control arm and to determine the optimal timing of vaccinations and any role of vaccine adjuvants.

Protecting immunocompromised patients against vaccine-preventable infectious disease is an opportunity to prevent morbidity and perhaps mortality that is frequently missed [83,84]. The current standard of care for patients with MGUS or SMM is observation, and during this time, it is important to remember routine vaccination against common infectious diseases during routine follow-up care, so-called ‘watch, wait, and vaccinate’.

## 6. Cardiovascular Disease in MGUS

Cardiovascular Disease (CVD) comprises arterial (coronary, peripheral, and cerebrovascular), venous thromboembolic and structural myocardial disorders. A retrospective cohort study matched for age and sex found that patients with non-IgM MGUS experienced an increased risk of arterial diseases compared to controls, but a significantly lower risk compared to patients with MM. The Hazard Ratio (HR) for arterial disease at five years was 1.3 (95% CI (confidence interval), 1.2–1.4) [85]. This increase in risk was independent of M-protein concentration and did not portend progression. In support of this observation, the mortality rate from CVD in MGUS patients has been repeatedly reported to be increased compared to matched controls without MGUS [8,14]. Venous thromboembolic (VTE) disease is also encountered more frequently in MGUS compared to controls HR 2.1 (95% CI, 1.7–2.5) [85]. The role of M-protein concentration is unclear with studies, each suggesting a positive, negative, and null correlation with risk of VTE [86,87,88,89]. There is no trial evidence to direct if these observations require specific management. For arterial disease, many patients would likely be recommended statin as primary prevention on the basis of age and male sex, but the role of aspirin remains controversial and has not been examined in this context. The rate of venous thromboembolism remains low (<1% per year in absolute terms [85]), and long-term anticoagulation carries a risk of major hemorrhage. For example, the Apixaban for the Prevention of Venous Thromboembolism in High-Risk Ambulatory patients (AVERT) trial of primary prevention of VTE in cancer patients with a low dose of the direct oral anticoagulant, apixaban, revealed a hazard ratio of 1.9 (95% CI, 0.39–9.24) compared to the placebo for major hemorrhage [90]. Therefore, if the risk of VTE were to be mitigated in MGUS, a different approach from prophylactic anticoagulation would likely need to be taken to provide a net benefit to the patient rather than exchange the risk of thrombosis for a similar risk of hemorrhage.

Our understanding of the pathogenesis of increased CVD risk in MGUS remains limited. Elevated levels of Factor VIII and the Von Willebrand factor have been found to be increased in MGUS cases, which may contribute to the increased risk of arterial thrombosis among MGUS patients [91]. More recently, the phenomenon of clonal hematopoiesis of indeterminate potential (CHIP) has been implicated mechanistically in atherosclerotic arterial disease [92]. CHIP is characterized by the acquisition of somatic mutations in preleukemic driver genes in hematopoietic stem cells [93]. Murine models provide substantial evidence that a major mechanism of increased cardiovascular risk in the context of CHIP is accelerated atherogenesis driven by inflammasome-mediated endothelial injury due to proinflammatory interactions between the endothelium and monocyte-derived macrophages [94]. Whether similar inflammatory mechanisms are responsible for the increased CVD risk in MGUS remains a subject of investigation. Potentially clonal hematopoiesis may co-exist with MGUS [95]. Recent studies on the presence of CHIP in MM report a prevalence of 22% [96]. Whether this extends to MGUS and correlates with the risk of CVD is not known. Overall, there is a growing appreciation of the increased risk of CVD in MGUS. However, a better understanding of the pathophysiological basis of the clinical observations is needed, together with trials of mitigating therapies in order to inform and guide clinical care of patients with MGUS. A summary of the key features of the co-morbidities associated with MGUS are summarized in Table 2.

## 7. Conclusions

Up to 90% of cases of MGUS are undiagnosed and are considered to predate symptomatic MM or lymphoproliferative disease by up to twenty years [97]. Therefore, MGUS as a contributory factor to the risk of fractures, nephropathy, renal impairment, and cardiovascular risk may be under-appreciated. However, the identification of MGUS in these scenarios may be coincidental and furthermore, not have an influence over management. Despite the potential anxiety caused by observation in the clinic, such monitoring has been associated with improved outcomes as it provides the opportunity to reinforce routine vaccination and bone protection strategies as well as for early identification of progression to symptomatic myeloma or lymphoproliferative disease [98,99]. There currently exist two prospective screening initiatives: The Iceland Screens Treats and Prevents Multiple Myeloma (iSTOPMM) study in Iceland [100], and the Predicting Progression of Developing Myeloma in a High-Risk Screened Population (PROMISE) study in the United States [101]. The iSTOPMM study screens individuals who are at least 45 years old for MGUS followed by the randomization and evaluation of intensive follow-up versus standard follow-up. The PROMISE study screens for MGUS and SMM among African Americans and first-degree relatives of MM individuals, who are at least 40 years of age, and prospectively follows them to determine clinical, immune, and genomic predictors of progression to MM. Such prospective cohorts may provide opportunities to address outstanding questions for supportive care in MGUS, as well as to improve our understanding of the pathophysiology of monoclonal gammopathies.

The concept of comorbidity in the context of ‘asymptomatic’ precursors seems paradoxical. However, the disease processes are varied and represent a spectrum of pathology. Medical science, and to a broader extent, human nature, tends to categorize continuous variables to facilitate comprehension of biology. However, such artificial categories may create barriers to our understanding of the pathology as a whole. The challenge for healthcare providers is to use these clinical definitions and criteria as an aid rather than a hindrance to appropriate care for patients. This review highlights the need to keep in mind that there is a need to manage the risks of patients as individuals rather than be bound by artificial criteria that may provide false reassurance. While it is imperative to ‘first do no harm’ and avoid the dangers of unnecessary therapy, a concerted effort is required to identify those for whom the benefits from therapies may improve quality of life, separate from an absence of malignant disease.

## Figures and Tables

**Figure 1 cancers-12-01554-f001:**
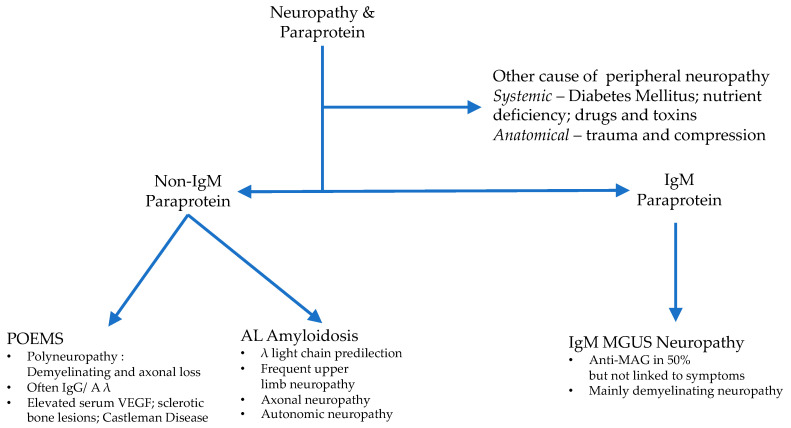
Initial clinical evaluation of a patient with a MGUS and peripheral neuropathy. MGUS = Monoclonal Gammopathy of Undetermined Significance; POEMS = Polyneuropathy, Organomegaly, Endocrinopathy, Monoclonal gammopathy, and Skin changes; VEGF = Vascular Endothelial Growth Factor; AL = Amyloid Light chain; MAG = myelin-associated glycoprotein).

**Table 1 cancers-12-01554-t001:** Summary of diagnostic criteria for monoclonal protein disorders.

Condition	Definition
Non-IgM MGUS	1. Serum monoclonal immunoglobulin ≤3 g/dL 2. Plasma cells in the bone marrow ≤10% 3. Absence of: lytic bone lesions, anemia, hypercalcemia, and renal impairment.
IgM MGUS	1. Serum monoclonal immunoglobulin ≤3 g/dL 2. Lymphoplasmacytic cells in the bone marrow ≤10% 3. Absence of: constitutional symptoms or symptoms and signs of hyper-viscosity, anemia or lymphadenopathy
Light chain MGUS	1. Abnormal free light chain ratio 2. Increased concentration of involved light chain 3. Complete loss of heavy chain immunoglobulin expression

Definitions derived from WHO. Classification of Tumors of Hematopoietic and Lymphoid Tissues, 2016. MGUS = Monoclonal Gammopathy of Undetermined Significance, WHO = World Health Organization.

**Table 2 cancers-12-01554-t002:** Summary of the presenting features and principles of management of co-morbidities associated with MGUS.

Co-Morbidity	Presentation	Management
Increased fracture risk	Increased incidence of reduced bone mineral density and atraumatic fractures compared to controls	Careful evaluation of myeloma-defining event Calcium/Vitamin D with bisphosphonates
Renal impairment (Monoclonal Gammopathy of Renal significance—MGRS)	Regular monitoring for renal impairment	Careful evaluation of myeloma-defining event. Prompt referral to nephrology for consideration of biopsy, considering the wide differential diagnosis
Peripheral neuropathy	Usually polyneuropathy λ light chain—consider POEMS and AL Amyloid	Close collaboration with neurology for testing, including nerve conduction studies and biopsy. Anti-MAG, ganglioside or asialo-GM1 antibodies may be useful to monitor response to therapy
Secondary immunodeficiency	Recurrent bacterial infections, especially sinopulmonary	Measure antibody levels for IgG, IgA and IgM as well as specific responses to tetanus, pneumococcus and Hemophilus. Consider prophylactic antibiotics, immunoglobulin replacement and ensure vaccinations maintained
Cardiovascular disease	Increased risk of arterial and venous thrombotic events	Encourage adherence to good practice in primary prevention—lifestyle modification and blood pressure control. No indication for prophylactic antiplatelet or anticoagulant therapy

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
