# Peer review of "Monoclonal Gammopathy of Undetermined Significance (MGUS)—Not So Asymptomatic after All"

_cancers, 2020, doi:10.3390/cancers12061554_

Round 1
Reviewer 1 Report
The authors present a comprehensive review about the main comorbidities associated with MGUS.
Based on the described evidence, I think it would be useful to add a summary with the recommendations for the practical management of the comorbidities analyzed in the paper.
I would also review Table #1 since:
- Hyperviscosity, anemia and lymphadenopathies are not usually considered constitutional symptoms
- Unsure if the criteria "increase in loss of involved light chain" is correct.
Author Response
Thank you for the time taken to review our manuscript. I shall address the points raised, below.
1. A tabular summary of the presentation and management principles now included as Table 2 in a self-contained section prior to the conclusions.
2. Table 1. I have corrected the typographical mistake 'increase in loss of involved light chain' to 'Increase of involved light chain'. I have altered the wording around constitutional symptoms and other symptoms and signs of disease due to IgM paraprotein disorders.
Thank you again for consideration of our manuscript.

Reviewer 2 Report
The authors are internationally respected academic clinicians and researchers in the realms of MGUS, SMM and MM. Their article follows a number of primary research and review publications that have arisen in last 4-5 years that are making the case that MGUS, whilst originally described as an asymptomatic pre-malignant state, is associated in a significant number of cases with co-morbidities associated with MM. The underlying processes driving these co-morbidities appear similar if not the same as in MM. Thus, although not ground breakingly original in all aspects, the article is a very valuable review of the literature and the most precise summary of where our understanding is on these issues with respect to fracture risk, renal impairment, peripheral neuropathy, secondary immunodeficiency and cardiovascular disease.
The article is very well written and makes a number of key observations and recommendations for the future. The authors reflections that 'Medical science,
and to a broader extent human nature, tends to categorize continuous variables to facilitate
comprehension of biology' and that these artificial categories may in fact 'create barriers to our understanding of the pathology as a whole' are important, and they discuss the need for clinical definitions and criteria surrounding MGUS to be used as an aid rather than a hindrance to appropriate care for patients.
Although the authors recognise that up to 90% of MGUS cases are undiagnosed and describe some of the issues about the challenges of not generating potential anxiety by introducing broad screening programmes, the article does not clearly state that the vast majority if not all MM arises from MGUS. Nor does it discuss that the majority of MM reaches diagnosis without prior recognition of MGUS. This would indicate that there may be an of excess fracture risk, renal impairment, peripheral neuropathy, secondary immunodeficiency and cardiovascular disease relating to MGUS than is currently recognised. The article would be improved if a discussion of this possibility was included.
Author Response
Thank you for the time taken to review our manuscript. We appreciate the extra points raised over the possible undiagnosed cases of MGUS that may contribute to the pathologies described in the manuscript.
I have supplemented the introduction with the disparity between MGUS and MM diagnoses - representing a 'clinical iceberg' of under diagnosed MGUS. The manuscript may therefore be used in two ways, to aid the hemato-oncologist with identifying symptoms attributable to MGUS and conversely professionals in other specialties who may under-appreciated the role MGUS may play in symptoms and signs presented to them. In the conclusion, in the first few lines, I have now added the point that the contribution of MGUS may be under-appreciated. Overall these points do show the diagnostic difficulty involved in how common ailments may coincide with clinical abnormalities such as MGUS, without necessarily a causative link. This is part of the challenge of this clinical question.
Thank you again for engaging in the topic and your consideration of our manuscript.

Reviewer 3 Report
The review "MGUS - not so asymptomatic after all" by OC.Lomas,....IM.Ghobrial is a summary of possible symptoms and future ways to approach patients with MGUS, which is in various aspects - as explained below -
- too long,
- very complicated, as written,
- entirely dogmatic in its implication to attest MGUS more pathology than needed and
- very unfortunately - not at all balanced in its presumption, that MGUS/SMM should be differently termed and possibly treated.
A much more evidence-based and -guided, critical review that catches both "MGUS-pro activity groups" and those "who consider this entity much lesser activity/action-driven in calling those with MGUS 'symptomatic or treatable'", would greatly help to make this article much more enjoyable to read and cite.
Major comments:
- Page 2, 1.paragraph: ".... the moderate symptoms......, MGUS represents an under-appreciated pathological state with significant morbidity and mortality."
There are many issues entirely incorrect in this one sentence, which needs to be either rephrased, or better taken off entirely from this paper, because a nuisance to read, i.e.
- a) MGUS-symptoms are normally not at all present, and most patients are accidentally found to have MGUS, or because they are screened (not an EMN- or IMWG-recommendation!) and diagnosed with a monoclonal gammopathy.
- b) "pathological state" implies again the above, which is incorrect, there is neither a "state" nor a "pathological status", rather than in most patients a paraprotein, that causes no symptoms whatsoever and will not necessarily lead a person to transform to either SMM or MM in their life-time (in 99% / year in MGUS no transformation to MM occurs).
- c) "under-appreciate": is also a term that might imply that screening of MGUS personal should be undertaken, which may occur in selected scientific projects, but is nothing that is recommended to be performed on all that are >50, >60, >70 or even >80 years of age, but no symptoms.
This is very important to state, because MGUS-screening does increase and leads indeed to substantial anxiety, unnecessary testing, unhappy discussions and endless diagnostics being performed with many numbers needed to perceive pathological findings and many falsely positive results being generated.
- d) What is the "high morbidity and mortality" in MGUS?, which sounds like "phrasing the extreme", where there is normally none and - although the authors imply this differently, most MGUS patients have a similar life expectancy than those without MGUS and happily live for many years without knowing of this precursor entity, until MM eventually may evolve and the initial diagnosis of a symptomatic disease being made.
- Table 1 is not really significant in its meaning and could be replaced by a much more meaningful figure or table. Moreover, it is less well understandable, what the authors mean with
- a) "increase in loss of involved light chain", rather than increase in involved light chains, i.e. kappa or lambda, and possibly decrease of the uninvolved light-chain, and
- b) "complete loss of heavy chain immunoglobulin expression", because i.e. if the patient has kappa-MGUS, then IgG, IgA and IgM levels will be classically not "entirely lost or suppressed", but are 'normally' or very often uncompromised.
- Page 3: The difficulty with all registry and population-based studies is, that they greatly lack details and are therefore very inaccurate, therefore all the relevant clinical details that are really of interest, are being missed and cannot be solved, therefore, the authors should be extremely careful of not overestimating their findings and more critically discuss their (and specifically these registry/data bank) findings.
- Page 4, 3.paragraph: That "a solitary plasmocytoma may harbor a plasma cell clone responsible for nephropathy,.... that radiotherapy may be curative,.... and limit renal impairment"
-> needs to be taken out of the manuscript, because
- entirely rare,
- even with local therapy of a plasmocytoma, cure may not be obtained and
- that renal impairment is entirely the cause in this scenario, often not the case.
- Moreover, reference 39, that is given for this unusual statement is from BLOOD 2006, thus now 14 years old.
- Page 5, 3.paragraph: The much more often involved reasons for peripheral neuropathy (PNP) is diabetes and alcohol abuse, which for the latter reason is not even mentioned, but much more important and prevalent than the possibly MGUS-associated PNP.
- The Fig.1 is also much less meticulously prepared than could be done and is really telling the reader what novel finding exactly? Could therefore be much improved.
- Page 6, 2.paragraph: POEMS is insufficiently described because not only the 5 acronyms stand for this and PNP and MGUS suffice, rather than major and/or minor criteria have to be determined and at least 3/5, rather than only 2 of these 5 need to be present to call POEMS securely as this.
- Page 6, 3. paragraph: The sentence "All forms of MGUS can progress to AL-amyloidosis, although l-LCs are almost always involved" is again incorrect and problematic, because:
- in AL-Amyloidosis, most often a small monoclonal clone, namely MGUS-appearance is typical
- l-LC are predominant in AL-amyloidosis, but with a frequency of 4:1 for l vs. k, respectively and
- why "although"?
- Page 6, paragraph 3: The therapy in POEMS and AL-amyloidosis is that of MM, but with much attenuated doses and more carefully application of multi-agent combinations, because often much lesser well-tolerated and because these patients are often frail.
What do the authors mean with "Therapy involves chemoradiotherapy and often autologous stem cell transplantation"?, because this is both inaccurate and for ASCT incorrect: due to the reasons above, unfortunately, often ASCT, both in POEMS and AL-amyloidosis, is not very often possible to perform (in amyloidosis in only approx. 5-10% of patients).
- Page 7, 2.paragraph: What do the authors mean with: "Indeed, in SMM, polyclonal hypogammaglobinemia had a median rate of progression of 68% at 5 years compared to 38% with monoclonal hypogammaglobinemia"?
- Page 7, last paragraph: "Therefore an approach to take during routine care is..... to identify those with deficient antibody responses to vaccines....."
-> this alludes that this would be routine care, which is certainly not the case, therefore, take out or rephrase.
- Page 8, 1. paragraph: "...during routine follow-up care, so that we 'watch, wait and vaccinate'."
-> This sentence is a presumption and suggestive, as if this was new or an innovative suggestion of the authors, which is entirely incorrect, because this is long known and done, thus rephrasing that this is a routine recommendation and done in every knowledgeable MM/SMM/MGUS/Amyloidosis-center, is needed.
- Page 8: Entire paragraph to cardiovascular disease and MGUS
-> should be taken out, because entire speculation, based on very weak data and publications and most likely absolute overestimated by these and those CVD-paper authors.
- Page 8, last paragraph: "Despite the potential anxiety (apparent in all MGUS persons!),... such monitoring has been associated with improved outcome...." -> really?
This is neither referenced nor correct, but again the presumption of the authors, who think that this might be the case, which is not correct.
Therefore, the authors would be well advised, if they much more self-critically discussed, whether MGUS screening is really "always advantageous".
- "Early identification of disease progression" has also not been shown to be the case, albeit Dr. Angela Dispenzieri has speculated that all the MM-progress we are seeing (in MM!, not MGUS) might in part be related to patients more early being diagnosed and treated, whereas formerly, MM patients were diagnosed with far more advanced disease and therefore might have "lived shorter with the MM disease".
For MGUS, however, this statement on "early identification benefits" is incorrect to make and not at all based on facts.
- Page 9, 2. paragraph: what is meant with "The concept of comorbidity in the context of "asymptomatic" precursors seems paradoxical"?
- Page 9, last paragraph: Entire paragraph should be rephrased or taken out and the authors should concentrate on a) doing no harm,
- b) not diagnosing and treating too early,
- c) not err on MGUS patients, not needing any interventions and
- d) to avoid treatment and SAEs in MGUS-persons of no necessity to undergo diagnostics, interventions, endless controls and even treatment or clinical studies.
This a) - d) needs excellent, careful expert advice and prudent MGUS/SMM/MM care, which the authors will also like to offer their and all worldwide patients, specifically those with asymptomatic MGUS, that are for the latter the typical rather than rare crowd.
Author Response
Thank you for your consideration of out manuscript. We examine how the care of individuals with MGUS is not entirely related to their low risk of progression to myeloma or lymphoproliferative disease. Specifically we look at the associated comorbidities of increased fracture risk, peripheral neuropathy, renal impairment, secondary immunodeficiency and cardiovascular disease.
General Points
In response to : 'very unfortunately - not at all balanced in its presumption, that MGUS/SMM should be differently termed and possibly treated.' MGUS and SMM are defined entities by WHO classification and each possess an International Classification of Diseases (ICD-10) code. We do not discuss progression from MGUS to smoldering myeloma to myeloma in this manuscript. Indeed the comorbidities described are distinct from myeloma defining events.
Specific Points
- We have altered the sentence identified by the reviewer to emphasise further the low burden of comorbidity that has been identified in numerous observational studies, published in significant medical journals and referenced throughout the manuscript. The last three sentences at the end of the first paragraph read as follows :
These co-morbidities afflict only a minority of patients with MGUS, but nevertheless they need to be considered on an individual patient basis. Since the prevalence of MGUS is much higher than that of MM, these more moderate symptoms that may be associated with MGUS may nevertheless have a significant impact on healthcare services. MGUS may represent an under-appreciated pathological state with attributable morbidity and mortality different from progression to malignancy.
a) We do not assert that there is a screening programme for MGUS nor do we advocate one. We identify the very low rate of progression to malignancy and the statistics quoted do indeed demonstrate that such symptoms affect a small minority of patients.
b) "pathological status" is not a quotation from this manuscript.
c) At no point do we advocate a screening program for MGUS, apart from the prospective observational studies that the reviewer acknowledges 'may occur in selected scientific projects'.
d) The text of the introduction has been altered to emphasise further the uncommon but still present nature of these co-morbidities. - a) (also labelled with point 3) The typographical error has been amended in the table.
b) (also labelled with point 4) We are referring in this part of the table to the light-chain variant of MGUS shown in the left hand column of that table. We reference the WHO Classification for Tumours of Haematopoietic and Lymphoid Tissues and use their phrasing accordingly - as shown in the footnote to the table. - We appreciate the limitations of retrospective studies and indeed we make the point that prospective studies, both observational or interventional, would help in this field.
- We appreciate the rarity of the situation and hence use the conditional 'may'. 'Moreover, reference 39, that is given for this unusual statement is from BLOOD 2006, thus now 14 years old.' We do not regard the age of an observation or reference as the sole indicator of its validity or significance.
- The first two sentences of the section demonstrate the broad range of aetiologies (beyond diabetes mellitus or alcohol misuse) that may cause peripheral neuropathy. The first sentence of the final paragraph of the section on neuropathy clearly states the care that needs to be taken in attributing causality to the finding of MGUS in association with a neuropathy.
- Figure 1 acts as schema to aid the non-specialist to approach the peripheral neurological manifestations that may be associated with monoclonal gammopathies. We have taken the opportunity to expand points further to expand upon the other causes of peripheral neuropathy. Careful attention to the schema does highlight the importance of excluding other causes of peripheral neuropathy. This may help in your concerns raised in point 5.
- A reference to POEMS syndrome and its diagnostic criteria are included in the text. Line 214 of the updated manuscript.
- Lines 218 and 219 shows updated syntax and word choice to emphasise the predominant involvement of lambda light chains in AL amyloidosis.
- A reference (61 in the updated manuscript) to a current description of the diagnosis and management of POEMS is included in the text, which explains the role of autologous stem cell transplantation.
10. We include 'to multiple myeloma' so the sentence now reads 'Indeed, in SMM, polyclonal hypogammaglobulinemia had a median rate of progression to myeloma of 68% at five years compared to 38% with monoclonal hypogammaglobulinemia'.
11. The use of 'routine' as been removed and the sentence referenced accordingly
12. The reviewer suggests routine vaccination is 'done in every knowledgeable MM/SMM/MGUS/Amyloidosis-center'. May we refer the reviewer to the previous sentence which highlights how such straightforward medical care is missed in practice. We have added a further reference to support this case.
13. We present the published evidence of the emerging topic of cardiovascular risk in MGUS. We acknowledge the limitations of our current understanding. We do not see this as a reason to exclude it from the discussion in its entirety.
14. The assertion of patient benefit from follow-up in MGUS is already referenced accordingly (Bianchi G et al Blood 2010). We do not advocate screening and "always advantageous" is not a phrase used by the authors of this manuscript and we are not clear why it is quoted as such.
15. We support the prompt treatment of symptomatic multiple myeloma according to current IMWG criteria. We do not advocate treatment at a precursor stage to prevent progression to myeloma as a standard of care. To make this clearer, the sentence has been rephrased from 'Early identification of disease progression' to 'Early identification of progression to symptomatic myeloma or lymphoproliferative disease'.
16. The sentence directly addresses the premise that MGUS is entirely asymptomatic and that the use of 'asymptomatic' is solely in relation to progression to myeloma or lymphoma. We present evidence through the review to advocate that patients benefit from holistic medical care that is not solely centred about progression to malignancy.
17. We are cognisant of the need to avoid over-treatment as by our assertion of the medical principle of 'do no harm'. The reviewer refers to 'early therapy', which is explicitly not the argument made in this manuscript. The final sentence clearly states that unnecessary therapy is to be avoided, but there are opportunities to improve quality of life for individuals more than simply comfirming the absence of malignant disease.
We hope that the points made above address your concerns.

Reviewer 4 Report
This is a very well written and comprehensive review. Indeed, the topic is very interesting, since MGUS is often referred as asymptomatic , whereas reports suggest that it can be associated to many pathological conditions.
A few minor comments:
- A table summarising the conditions associated to MGUS described in this review, with their management strategy (suggested diagnostic work-up and possible treatments) could be helpful for the reader.
- Paragraph 2: please double-check the references, since reference 22, 23, 27 etc seem incorrect (seem related to the following paragraph).
- Paragraph 2: incidence of MGUS increases in the elderly population, as well as the incidence of osteoporosis. Do the authors think that increased incidence of low bone mineral density observed in MGUS might be the consequence of the increased incidence of MGUS in the elderly?
- A brief paragraph about cutaneous manifestations associated to MGUS could be of interest for the reader (see for example: Lipsker D. Monoclonal gammopathy of cutaneous significance: review of a relevant concept. J Eur Acad Dermatol Venereol JEADV. 2017;31:45–52)
Author Response
Thank you for the time taken to review the manuscript and the comments provided. We shall address your points below.
1. A tabular summary of the presentation and and management of the associated conditions discussed int he manuscript has been formed in the penultimate section of the manuscript.
2. The references have been corrected accordingly.
3. We appreciate the coincidence of osteoporosis with age and MGUS. However, we feel it is a minor contributor that does not correlate with the sex distribution of low bone mineral density as a whole. We were also interested by the possibility that volumetric changes could underlie the increased fracture risk rather than reduced bone mineral density (Thorsteinsdottir et al 2017) and therefore BMD may not be the sole determinant of risk stratification in the MGUS cohort.
4. We have made reference to cutaneous manifestations in our introduction with a link to the excellent review suggested.
Thank you again for your comments and suggestions to improve the manuscript.

Round 2
Reviewer 3 Report
Improved, although I still do not share the authors' statements and plenty of their thoughts and beliefs.